# Profiling RT-LAMP tolerance of sequence variation for SARS-CoV-2 RNA detection

**Esta Tamanaha**[ID]**, Yinhua Zhang, Nathan A. Tanner**[ID] *

Research Department, New England Biolabs, Ipswich, Massachusetts, United States of America

* tanner@neb.com

## Abstract

The ongoing SARS-CoV-2 pandemic has necessitated a dramatic increase in our ability to conduct molecular diagnostic tests, as accurate detection of the virus is critical in preventing its spread. However, SARS-CoV-2 variants continue to emerge, with each new variant potentially affecting widely-used nucleic acid amplification diagnostic tests. RT-LAMP has been adopted as a quick, inexpensive diagnostic alternative to RT-qPCR, but as a newer method, has not been studied as thoroughly. Here we interrogate the effect of SARS-CoV-2 sequence mutations on RT-LAMP amplification, creating 523 single point mutation "variants" covering every position of the LAMP primers in 3 SARS-CoV-2 assays and analyzing their effects with over 4,500 RT-LAMP reactions. Remarkably, we observed only minimal effects on amplification speed and no effect on detection sensitivity at positions equivalent to those that significantly impact RT-qPCR assays. We also created primer sets targeting a specific short deletion and observed that LAMP is able to amplify even with a primer containing multiple consecutive mismatched bases, albeit with reduced speed and sensitivity. This highlights RT-LAMP as a robust technique for viral RNA detection that can tolerate most mutations in the primer regions. Additionally, where variant discrimination is desired, we describe the use of molecular beacons to sensitively distinguish and identify variant RNA sequences carrying short deletions. Together these data add to the growing body of knowledge on the utility of RT-LAMP and increase its potential to further our ability to conduct molecular diagnostic tests outside of the traditional clinical laboratory environment.

## Introduction

Molecular diagnostic techniques are able to provide definitive identification of infectious agents through specific detection of DNA or RNA sequences of interest. However, target pathogens, and particularly RNA viruses, naturally accumulate mutations and changes in their genomic sequences that can impact the sensitivity and accuracy of the molecular diagnostic tests when the mutations occur in the regions targeted by the oligonucleotide primers and/or probes. The ongoing SARS-CoV-2 pandemic has seen the emergence of numerous viral variants from different regions of the world, with prominent effects on detection using molecular assays [1–7]. For example, the B.1.1.7 "alpha" variant notably featured a 6-base deletion

**Data Availability Statement:** All relevant data are within the paper and its Supporting information files.

**Funding:** New England Biolabs (www.neb.com) has funded this study. All authors (ET, YZ, NT) are

employees and shareholders of New England Biolabs, manufacturer of LAMP reagents described in the manuscript.

**Competing interests:** New England Biolabs (www. neb.com) has funded this study. All authors (ET, YZ, NT) are employees and shareholders of New England Biolabs, manufacturer of LAMP reagents described in the manuscript. This does not alter our adherence to PLOS ONE policies on sharing data and materials.

(removing 2 amino acids of the spike protein, Δ69–70) which causes a failure of the S Gene assay in the widely used TaqPath® COVID-19 multiplex RT-qPCR test [8]. The other targets in this assay are unaffected, resulting in the ability to provide preliminary identification of variant RNA during detection, identifying a sample for further sequencing analysis and variant determination. In this particular case the other targets can be used for diagnostic detection, but the sensitivity of RT-qPCR to variant mutations is a significant concern to diagnostic testing given the worldwide reliance on the method. Early in 2021 the United States Food and Drug Administration (FDA) issued guidance to molecular test developers [9] calling for evaluation of variant sequences in development of any molecular test, emphasizing the significance of understanding assay performance in the presence of targeted region mutations.

The effects of mutations on qPCR assays are well-studied, with even single-base mutations having generally deleterious effect [1–8, 10, 11]. Likewise, single-base changes from SARS-CoV-2 mutations have been demonstrated to have significant impact on RT-qPCR assays. One N gene mutation (G29140U) falling in a Forward PCR primer resulted in a 5–6 $C_q$ increase with an estimated 67-fold drop in assay sensitivity [12]. Another N gene single base change (G29179U) in the CDC N2 Forward primer produced a ~4 $C_q$ increase in PCR and a ~5-fold decrease in sensitivity [13]. A study of 14 SARS-CoV-2 RT-qPCR assays with mismatches identified several primer positions as particularly sensitive to mutation, with up to a 7.6 Cq increase compared to fully base-paired templates [7].

In contrast, loop-mediated isothermal amplification (LAMP) diagnostic assays are a more recent technology and accordingly we have comparatively less information on mutation effects. RT-LAMP has played a role in point-of-care and fieldable diagnostics, but during the SARS-CoV-2 pandemic RT-LAMP has emerged as a prominent and widely-used molecular diagnostic method [14]. Isothermal amplification can be performed with much simpler instrumentation than PCR, and detection of RT-LAMP can be conducted directly by visual color change [15, 16], fluorescence [17], sequence-specific probes such as DARQ [18] and molecular beacons [19], or coupled to secondary molecular analysis platforms such as CRISPR [20, 21] and next generation sequencing (e.g. LamPORE and LAMP-Seq) [22, 23]. Several SARS-CoV-2 diagnostic protocols based on RT-LAMP are currently being used for large-scale tests [20, 21, 24, 25] and can utilize crude or unpurified samples, saving time and cost while increasing testing flexibility and portability, exemplified by the utilization of RT-LAMP as the first molecular diagnostic for at-home use granted EUA in November 2020 [25].

As with qPCR, LAMP relies on oligonucleotide primers and mutations in the targeted regions may affect amplification efficiency. However, there is significantly more complexity in LAMP, as it utilizes 6 primers derived from 8 regions in the target sequence, with each of these regions similar in length to a PCR primer [26, 27]. These 6 primers perform different roles during the amplification: F3 and B3 are located outside of the targeted region to aid in release of initial amplicons via strand displacement and are not incorporated into the amplification products. FIP and BIP are composed of two distinct target sequences and are the core primers responsible for the generation of the LAMP hairpin "dumbbell structure" and subsequent exponential amplification. LoopF and LoopB initiate from regions formed by "looping" from FIP and BIP and serve to further increase the speed of amplification.

Previous LAMP assays have been designed specifically to target single nucleotide polymorphisms (SNPs) in a sequence by placing the mutation directly at the 3′ or 5′ terminal base of both the FIP and BIP primers [28]. However, this arrangement is intended to target a known mutation and would not occur in a standard LAMP diagnostic assay. Natural mutations would most frequently consist of single base changes at any location along the LAMP primers, or more rarely, multi-base deletions. For example, the B.1.1.7 Alpha variant, in addition to the 6-base deletion in the S gene described above, contains a 9-base deletion removing 3 amino

acids in Orf1a (3675–3677, the "SGF" deletion), and also >18 characteristic single base change mutations. The B.1.617.2 Delta variant contains an additional >20 different point mutations and 2 deletions while Omicron (B.1.1.529) contains >30 point mutations and 4 deletions unique from both the Alpha and Delta variants [29].

Emerging SARS-CoV-2 sequence changes can be monitored for potential overlap with amplification assay primers through the New England Biolabs Primer Monitor tool. Using this tool, mapping common RT-LAMP primers onto SARS-CoV-2 sequences from GISAID shows numerous locations of mutation in small numbers of isolates worldwide. Of higher concern are mutations seen with significant prevalence, (>10% of sequences, deposited from at least 2 geographic locations, independent from any of the well-known variants of interest or concern; see Materials and methods). In each case, only a single base change occurred within any of the primer set regions. In this study, we sought to comprehensively characterize the impact of mutations on RT-LAMP primers, and transferred the prevalent mutations to each position of all 18 oligonucleotide primers making up 3 SARS-CoV-2 RT-LAMP assays. As shown below, RT-LAMP displayed consistent performance in speed and sensitivity across all 523 mutational positions tested.

While this robustness is valuable for a reliable, variant-tolerant molecular diagnostic assay, there is additional value in assays that can distinguish and specifically identify particular variants with the amplification reaction. Sequencing of positive samples will remain the most powerful and sensitive method for variant identification, but molecular diagnostic assays can present a significantly faster and cheaper approach, if appropriately designed. The widely-used TaqMan probes rely on specific hybridization to targeted sequences and may display greater sensitivity to mutations than when they occur in primer regions; the Δ69–70 deletion occurs in the TaqPath probe region and completely prevents detection of RNA with that deletion. While only Alpha and Omicron carry the Δ69–70 deletion, all but Delta carry a 3-amino acid deletion at positions 3675–3677 (3674–3676 in Omicron) in the Orf1a sequence. This relatively large deletion provides a reliable means for designing molecular diagnostic assays that can distinguish between strains differing at this sequence location [8], and we sought to target that region with deletion-specific LAMP primers. While there was some impact on speed and sensitivity, even this significant stretch of mismatches did not inhibit LAMP completely and showed little effect at high copy numbers. Instead, we show below a molecular beacon strategy is needed for variant targeting using RT-LAMP.

## Materials and methods

### Single point mutation LAMP primers

Three previously described SARS-CoV-2 LAMP primer sets for SARS-CoV-2 were chosen to profile mutational position effects: As1e [30], E1, and N2 [16] (Table 1). Four point mutations in regions targeted by these primers were identified on some of these LAMP primers in GISAID sequences as monitored by NEB Primer Monitor (https://primer-montor.neb.com) in >10% of deposited sequences from >1 reporting location. These mutations were: C2395T, in the As1e BIP primer; G2416T, in the As1e LoopB primer; T29148C, in the N2 F3 primer; and G29179T, in the N2 LoopF primer (locations noted in Table 1 below). We then modeled these known mismatches (G:T, C:T, T:G, G:A) by introducing a mismatch at every base position for each of the primers, changing: C→T, T→C, G→T, or A→C. The resulting 541 primers, including wild-type primers for consistency, were synthesized in 96-well plates and resuspended at 10x concentration (2 μM F3, B3; 16 μM FIP, BIP; 4 μM LoopF, LoopB) by Integrated DNA Technologies (IDT) and spot-checked for concentration accuracy using 90 of the provided oligos.

**Table 1. LAMP primers.**

| Assay | Primer | Sequence |
|---|---|---|
| As1e | F3 | CGGTGGACAAATTGTCAC |
| | B3 | CTTCTCTGGATTTAACACACTT |
| | FIP | TCAGCACACAAAGCCAAAAATTTATTTTTCTGTGCAAAGGAAATTAAGGAG |
| | BIP | TATTGGTGGAGCTAAACTTAAAGCCTTTTCT*G*TACAATCCCTTTGAGTG |
| | | *Mutation*: *C2395U*, *EU/Alberta* |
| | LF | TTACAAGCTTAAAGAATGTCTGAACACT |
| | LB | TTGAATTTAGGTGAAACATTTGT*C*ACG |
| | | *Mutation*: *G2416U*, *California* |
| E1 | F3 | TGAGTACGAACTTATGTACTCAT |
| | B3 | TTCAGATTTTTAACACGAGAGT |
| | FIP | ACCACGAAAGCAAGAAAAAGAAGTTCGTTTCGGAAGAGACAG |
| | BIP | TTGCTAGTTACACTAGCCATCCTTAGGTTTTACAAGACTCACGT |
| | LF | CGCTATTAACTATTAACG |
| | LB | GCGCTTCGATTGTGTGCGT |
| N2 | F3 | ACCAGGAACTAA*T*CAGACAAG |
| | | *Mutation*: *U29148C*, *Brazil* |
| | B3 | GACTTGATCTTTGAAATTTGGATCT |
| | FIP | TTCCGAAGAACGCTGAAGCGGAACTGATTACAAACATTGGCC |
| | BIP | GCATTGGCATGGAAGTCACAATTTGATGGCACCTGTGTA |
| | LF | GGG*GG*CAAATTGTGCAATTTG |
| | | *Mutation*: *G29179U*, *Mexico* |
| | LB | CTTCGGGAACGTGGTTGACC |
| SGF-del | F3 | TTCTCTTGCCACTGTAGC |
| | B3 | AGTGTCCACACTCTCCTAG |
| | wt-FIP | CCAGACAAACTAGTATCAACCATTCTATATGCCTGCTAGTTGG |
| | Del-FIP | *CTT*CAAACTAGTATCAACCATTCTATATGCCTGCTAGTTGG |
| | | *Mutation*: *del3675-3677* |
| | BIP | GTTTTAAGCTAAAAGACTGTGTTATGGTTCTTGCTGTCATAAGGATT |
| | LF | CCAACCATGTCATAATACGCATC |
| | LB | ATGCATCAGCTGTAGTGTTACT |
| SGF-MB | F3 | GCTTTTGCAATGATGTTTGTC |
| | B3 | AGTGTCCACACTCTCCTAG |
| | FIP | CCAACTAGCAGGCATATAGACCATACATTTCTCTGTTTGTTTTTGTTACC |
| | BIP | ATGACATGGTTGGATATGGTTGGTTCTTGCTGTCATAAGGATT |
| | LF | AAGCTACAGTGGCAAGAGAA |
| | LB | ATGCATCAGCTGTAGTGTTACT |
| | MB-WT | /5Cy3/GGAGCTT+T+GT+CTGGTTT+TA+AG+CTCC/3IAbRQSp/ |
| | MB-DEL | /56-FAM/CGCAGTT+T+GAAG+CTAAA+A+GA+CTGCG/3IABkFQ/ |

## Deletion LAMP primers and molecular beacons

Two primer sets were designed targeting the SGF deletion. The first set (SGF-del) placed 3 deletion-specific bases at the 5' end of the FIP and compared to the FIP matching the wild-type sequence. The second set (SGF-MB) was designed to amplify both wild-type and SGF deletion for detection by molecular beacons. This set was designed with the SGF deletion located between B1c and LB using the NEB LAMP Primer Design Tool (https://lamp.neb.com) and with enough length between B1c and LB to accommodate the location of molecular

beacons in this region. Molecular beacons targeting either wild-type or the SGF deletion sequence were designed using principles according to [19]. As the deletion is located in a relatively AT-rich region, the annealing temperatures for these locked nucleic acid (LNA) beacons are lower: the calculated Tm of the annealed beacon-target for wt MB and SGFdel MB is 63.9 and 62.5°C, and the stem is 55.1 and 54.7°C, respectively. These beacons were synthesized as Affinity Plus qPCR Probes by IDT with sequences shown in Table 1.

### RT-LAMP reactions

RT-LAMP reactions were performed using WarmStart® LAMP Kit (DNA & RNA) (E1700) with standard primer concentrations (0.2 µM F3, 0.2 µM B3, 1.6 µM FIP, 1.6 µM BIP, 0.4 µM Loop F, 0.4 µM Loop B) in the presence of 40 mM guanidine hydrochloride [16] in 25 µL on 96-well plates at 65°C in a Bio-Rad CFX96 instrument. LAMP amplification was measured by including 1X NEB LAMP Dye (B1700) or 1 µM SYTO™-9 (ThermoFisher S34854), 0.5 µM SGFdel or 1.0 µM wt beacon, and fluorescent signal was acquired at 15 second intervals. Synthetic SARS-CoV-2 RNAs were obtained from Twist Bioscience (Control 2 for WT MN908947.3; Control 14 for B.1.1.7; Control 16 for B.1.351; and Control 17 for P.1) and diluted in 10 ng/µL Jurkat total RNA based on the copy number provided by the manufacturer.

## Results

### Positional mutation effects

To mimic the effect of a potential SARS-CoV-2 variant in an RT-LAMP assay, we focused on single point mutations at each primer base position and the SGF deletion that is found in several variants of concern. For the single point mutation primers, each of the 523 variant primers from the 3 assays (As1e, N2, E1) was tested in RT-LAMP reactions with three different SARS-CoV-2 RNA copy number concentrations: 100, 200, and 10,000 copies in order to gain a sense of the mutation effect on reaction speed and sensitivity. Both lower concentrations allowed for amplification effects to be confidently determined outside of stochastic performance when close to the limit of detection (LOD ~50 copies) in the 100 copy reactions, particularly for the As1e primer set which displays slightly lower sensitivity in our testing (S1 and S2 Figs). The reaction output speed was measured relative to the fully-complementary wild-type primer and plotted along the position in the primer sequence (Fig 1). Overall, the ~4,500 RT-LAMP assays

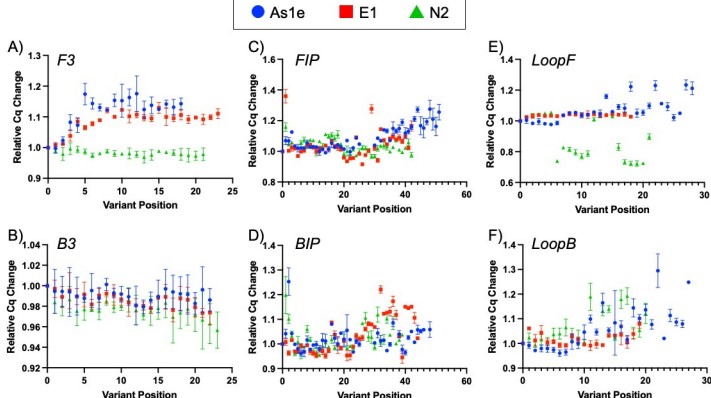

**Fig 1. Mutation position effects on RT-LAMP amplification.** Plots of the effects on amplification speed relative to the WT primer set for all 3 assays explored, As1e (blue circle), E1 (red square), N2 (green triangle). (A) F3 primer, (B) B3 primer, (C) FIP primer, (D) BIP primer, (E) Loop F primer, (F) Loop B primer.

containing single point mutations within any of the six primers resulted in minimal to no effect on the ability to amplify the target at any of the copy numbers used, regardless of primer and gene target. The most common result observed was a 5–10% reduction in amplification speed in the presence of a mismatch.

Evaluating the effects across the different primers revealed some minor differences based on position and role of the primer. The B3 primers showed remarkably little impact in any of the 69 variant primer sets for the 3 amplicons, though the F3 primer did see some slowing when mismatches were present away from the 5′ end in the E1 and As1e sets (Fig 1A and 1B). These primers are the least critical to the reaction, but the functional difference between the two may suggest differential mismatch tolerance during the reaction initiation step (B3 annealing to single-stranded RNA and extension by RTx reverse transcriptase) versus the strand invasion and displacement via *Bst* 2.0 polymerase that occurs for the F3 primer. With the more critical FIP and BIP primers, the 3′ half of the primer (F2/B2) serves to invade and prime double-stranded DNA, with the 5′ half annealing to displaced product strands to form the 'loop' dumbbell shapes for amplification. As shown in Fig 1C and 1D, more of the variant primer sets caused amplification delays relative to the fully base-paired control when the mutations were located toward the 3′ end of the FIP and BIP (F2/B2 regions) in all 3 LAMP assays. The extreme 5′ end displayed an increased mismatch effect on reaction speed, likely indicating an impact on polymerase extension from a mismatch in the looped-back LAMP hairpin structure, but detection sensitivity was not impacted even with those primer sets. Surprisingly, the speed of the N2 assay increased when mutations were present at internal and 3′ positions of the LoopF primer. This was not a consistent effect, and while difficult to anticipate from sequence prediction its underlying mechanism will be investigated further in future LAMP assays. As a summary of the effects, Table 2 lists the number of positions from each primer that resulted in a change of more than 10% in time to detection from the WT primer baseline. Though overall effects on amplification were minimal, the greater impact of 3′ mutations is clear from the trends in Fig 1. And while a significant number of variant primers resulted in decreased reaction speeds, in all 523 variant primers tested no mutation position prevented amplification in the RT-LAMP reaction with SARS-CoV-2 RNA even with lower RNA copy numbers.

## Effect of short deletions on RT-LAMP and detection by molecular beacons

We next looked to characterize the effects of short deletions on RT-LAMP, as have occurred in several SARS-CoV-2 assays. Systematically screening positional deletion effects, as done with single-base mutations above, would require the generation of hundreds of RNA templates. Instead, we took the existing, frequently occurring Orf1a SGF deletion and designed primer sets with that mutation at the 5′ end of the FIP (Table 1), the position of highest sensitivity to mismatches, as determined above. At the the early phase of LAMP amplification, this FIP

**Table 2. The effect of single point mutations on RT-LAMP performance.**

| Primer | No. Positions with >10% LAMP Time Change | Total Bases | Fraction Positions with >10% Effect |
|--------|------------------------------------------|-------------|-------------------------------------|
| F3 | 22 | 62 | 0.35 |
| B3 | 0 | 69 | 0 |
| FIP | 27 | 131 | 0.21 |
| BIP | 21 | 128 | 0.16 |
| LF | 18 | 68 | 0.26 |
| LB | 13 | 66 | 0.19 |

forms a mismatched 3' end with wild-type target in the dumbbell. This multi-base mismatch would then interfere with the amplification from the dumbbell and also subsequent priming for polymerization. Testing RT-LAMP containing this SGF-Del-FIP with low copy number (~10 copies), amplification with fully matched SGF deletion target (B.1.1.7 RNA) was efficient and sensitive while no amplification was observed with the mismatched wild-type target, indicating that mismatched FIP affected detection (S3 Fig). However, with higher copy numbers (>1000 copies), primer sets with mismatched target sequence were not completely discriminatory, with amplification of B.1.1.7 RNA and wild-type primers (and vice versa), though with reduced speed (S3 Fig). These results indicates that while some degree of LAMP impact was achieved by placing the SGF deletion at the 5′ end of the FIP, the effects on amplification were insufficient for confident and consistent variant identification with this strategy. And while robust to even 9-base deletions at the FIP end with moderate copy numbers, the significant impact in low copy reactions would indicate mutations of this severity at key positions and would affect the performance of a LAMP diagnostics assay.

Toward a more reliable variant discrimination strategy, we designed another primer set and detected the amplification with two molecular beacons targeting either the wild-type or the variant target region (Table 1). We initially evaluated this primer set for specificity and sensitivity using dsDNA binding as a reporter and found it was able to detect both WT and SGF variant RNA with similar sensitivity of approximately 50 copies and with no apparent non-template amplification signal in 40 minute reactions. We then evaluated detection using the molecular beacons with 10-fold dilutions of wild-type or B.1.1.7 synthetic RNA. Both molecular beacons detected their intended targets as designed with robust specificity and even at 10,000 copies of target RNA they recognized only their intended amplification target sequence (Fig 2).

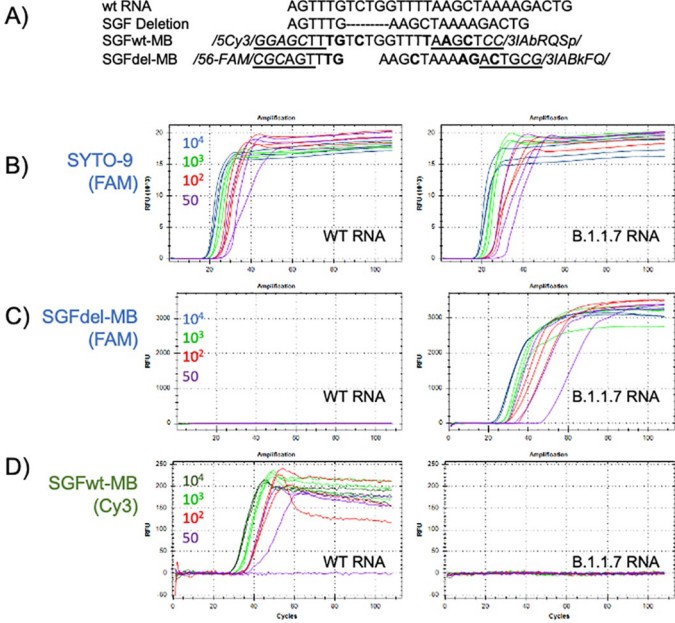

**Fig 2. Detection of target RNAs by SGFwt and SGFdel molecular beacons.** (A) Sequence comparison for wt, SGF deletion, SGFwt-MB and SGFdel-MB. Dashes: bases deleted in SGF deletion; Bold: LNA base; Underlined: stem region; Italics: non-target sequence, attached fluorophores and quenchers. LAMP reactions with either WT RNA (left panels) or B.1.1.7 RNA (right panels) in the presence of (B) SYTO-9, (C) SGFdel-MB, or (D) SGFwt-MB. The primer set amplifies both the wt and B.1.1.7 RNA with similar efficiency as detected with SYTO-9 (B). When beacon was added as a reporter, both SGFdel-MB (C) and SGFwt-MB (D) detected only their intended template RNAs from 50–10,000 copies.

**Table 3. Specific detection of variant RNA with LAMP and molecular beacons.**

|  | WT | B.1.1.7 | B.1.351 | P1 |
|---|---|---|---|---|
| SYTO-9 | 21 | 21 | 23 | 24 |
| Beacon | 18 | 17 | 24 | 23 |

Positives from 24 repeats, 50 copies/reaction

**Table 4. Dual-beacon RT-LAMP for variant RNA detection.**

|  | Single Beacon | | Duplex | |
|---|---|---|---|---|
|  | Cy3 (WT) | FAM (Del) | Cy3 (WT) | FAM (Del) |
| WT RNA | 20 | - | 20 | 0 |
| P1 RNA | - | 18 | 0 | 22 |

Positives from 24 repeats, 50 copies/reaction

Sensitivity of detection with the SGFdel beacon was evaluated with 3 different variant RNAs [B.1.1.7 (alpha), B.1.351 (beta), or P.1 (gamma)]. RT-LAMP reactions were carried out in 24 replicates with ~50 copies of synthetic RNA for each variant, and detected with either molecular beacon or dsDNA binding dye. Both detection approaches were able to detect all the variant RNAs with similar efficiency (Table 3). We further tested combining both SGFwt and SGFdel molecular beacons in the same LAMP reaction to determine the specificity of the molecular beacons for their specific sequence target. Results from these reactions showed that target RNAs were reliably detected with the same level of sensitivity when both beacons were present, and we observed no interference between the two beacons (Table 4) or detection of the incorrect RNA target sequence.

## Discussion

RT-LAMP has become a significant molecular diagnostic tool during the COVID-19 pandemic due to its simplicity and flexibility, expanding the reach of molecular methods beyond the clinical laboratory where RT-qPCR remains the dominant method. Building a greater understanding of how RT-LAMP assays perform will be critical to increasing their utility, and tolerance to mutations is a pressing need for this and any future viral detection effort. Here we established the first comprehensive screen of LAMP primer tolerance to mutation, investigating a single base mutation at every position of every primer in three prominent SARS-CoV-2 RT-LAMP assays. Remarkably, we find very little impact of the single base changes, with only marginal effect on amplification speed in most positions. In comparison, single-base changes in SARS-CoV-2 mutations have been demonstrated to have significant impact on RT-qPCR assays, shifting performance up to 7.6 Cq and reducing detection sensitivity 10–100 fold depending on the mutation and its position [7, 12]. Mutations at equivalent positions in LAMP primers showed minimal impact on speed and sensitivity.

Profiling effects at every LAMP primer position, we observed greater impact near the 3′ end of the FIP and BIP primers. This effect suggests the importance of initiation *via* the F2 and B2 regions in driving the formation of the LAMP dumbbell structures. However, while the speed decreases were more pronounced at those positions, the effect was minimal as compared to equivalent mutation positioning in PCR, where a 3′ mismatch can cause a >100-fold sensitivity effect and can be used to completely discriminate a SNP due to inefficient mismatch extension [10].

The robustness of RT-LAMP to sequence variation is a significant benefit over RT-qPCR, with reduced worry about deleterious effects and false negative results from the commonly emerging single-base changes that could occur with some frequency in the regions targeted by the LAMP primers. Additionally, many RT-LAMP assays combine primer sets for added speed and sensitivity [16], adding an additional layer of protection against possible sequence variation. Beyond the most common single-base changes, even targeting a 9-base deletion at the most sensitive location of the FIP primer did not result in complete amplification failure, as RT-LAMP was able to amplify from the significantly mismatched primer-template duplex at moderate copy numbers.

The converse of this assay robustness is an inherent difficulty in identifying variants relying on the amplification reaction. While sequencing offers greater confidence and detail for variant calling, the ability to utilize the diagnostic amplification for prospective variant identification as with the TaqPath S-gene dropout can be a valuable feature of potential diagnostic methods. As described above, we observed difficulty targeting the large 9-base SGF deletion by typical LAMP primer design alone. However, by utilizing a molecular beacon approach as first described by [31]. we were able to accurately amplify and identify RNA from the three SARS--CoV-2 variant sequences containing the SGF deletion. By combining the beacons for the wild-type and deletion sequence, we could call wild-type or variant based on the detected sequence, indicating the potential ability for variant calling in the RT-LAMP assay by multiplexed beacon design. Requiring a molecular beacon and fluorescence detection does of course remove some of LAMP's advantages in portability and simplicity, but in situations where variant calling is desired, LAMP can enable these applications, still with faster reaction times than PCR and on plate readers and simpler detection equipment that does not require thermal cycling.

Taken together these data position RT-LAMP as an attractive diagnostic method with a high level of tolerance to sequence mutations. Recent FDA guidance described a need for understanding this tolerance for any molecular diagnostic test, and use of RT-LAMP could convey increased confidence to developers that a test will maintain performance. In situations where variant identification is desired, the robustness of LAMP primers is a detriment, but use of molecular beacons provides a sensitive addition to specific sequence detection. RT-LAMP remains a promising molecular diagnostic method and by further characterizing its behaviors and increasing its applicability we hope to further enable it to bring diagnostic testing to field and point-of-care settings where its advantages can be more fully utilized.

## Supporting information

**S1 Fig. Mutation position effects on RT-LAMP amplification at 100 copies of SARS-CoV-2 RNA.** Plots of the effects of change relative to the WT primer set for all three assays explored, As1e (blue circle), E1 (red square), N2 (green triangle) at 100 copies of SARS-CoV-2 RNA. (A) F3 primer, (B) B3 primer, (C) FIP primer, (D) BIP primer, (E) Loop F primer, (F) Loop B primer.
(TIFF)

**S2 Fig. Mutation position effects on RT-LAMP amplification at 200 copies of SARS-CoV-2 RNA.** Plots of the effects of change relative to the WT primer set for all three assays explored, As1e (blue circle), E1 (red square), N2 (green triangle) at 200 copies of SARS-CoV-2 RNA. (A) F3 primer, (B) B3 primer, (C) FIP primer, (D) BIP primer, (E) Loop F primer, (F) Loop B primer.
(TIFF)

**S3 Fig. Discrimination of SGF deletion with mismatched primers.** (A) Comparison of sequences of the 5' of the FIP primers for detecting wt or SGF deletion. (B) LAMP amplification with primer set containing SGFdel-FIP. 8x repeats of LAMP reactions were performed with 1000 or 10 copies of RNA from B117 or wt. The speed of LAMP is shown as "# of Cq" with each Cq equal to 24s. Positive amplification threshold is Cq<60.
(TIFF)

**S1 Table.**
(XLSX)

## Acknowledgments

The authors thank Eric Hunt (NEB) for assistance designing the variant primer panels and Nicole Nichols (NEB) for inspiration to conduct the full set of variant profiling experiments. We are grateful to New England Biolabs for fostering an environment of scientific discussion and collaboration, without which this work would not have been possible.

## Author Contributions

**Conceptualization:** Nathan A. Tanner.

**Data curation:** Esta Tamanaha, Yinhua Zhang.

**Supervision:** Nathan A. Tanner.

**Writing – original draft:** Esta Tamanaha, Yinhua Zhang.

**Writing – review & editing:** Esta Tamanaha, Yinhua Zhang, Nathan A. Tanner.

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
