## [Decision Letter · Decision Letter 0]

7 Jan 2022

PONE-D-21-33500Profiling RT-LAMP tolerance of sequence variation for SARS-CoV-2 RNA detection

PLOS ONE

Dear Dr. Tanner,

Thank you for submitting your manuscript to PLOS ONE. After careful consideration, we feel that it has merit but does not fully meet PLOS ONE’s publication criteria as it currently stands. Therefore, we invite you to submit a revised version of the manuscript that addresses the points raised during the review process.

We look forward to receiving your revised manuscript.

Kind regards,

Grzegorz Woźniakowski, Full professor, PhD, ScD

Academic Editor

PLOS ONE

Journal Requirements:

"The authors thank Eric Hunt (NEB) for assistance designing the variant primer panels and Nicole Nichols (NEB) for inspiration to conduct the full set of variant profiling experiments. We are grateful to New England Biolabs for funding the work and fostering an environment of scientific discussion and collaboration, without which this work would not have been possible."

"New England Biolabs (www.neb.com) has funded this study. All authors (ET, YZ, NT) are employees and shareholders of New England Biolabs, manufacturer of LAMP reagents described in the manuscript"

Additionally, because some of your funding information pertains to [commercial funding//patents], we ask you to provide an updated Competing Interests statement, declaring all sources of commercial funding. 

In your Competing Interests statement, please confirm that your commercial funding does not alter your adherence to PLOS ONE Editorial policies and criteria by including the following statement: "This does not alter our adherence to PLOS ONE policies on sharing data and materials.” as detailed online in our guide for authors  http://journals.plos.org/plosone/s/competing-interests.  If this statement is not true and your adherence to PLOS policies on sharing data and materials is altered, please explain how. 

Please include the updated Competing Interests Statement and Funding Statement in your cover letter. We will change the online submission form on your behalf

"New England Biolabs (www.neb.com) has funded this study. All authors (ET, YZ, NT) are employees and shareholders of New England Biolabs, manufacturer of LAMP reagents described in the manuscript."

We note that you received funding from a commercial source: [Name of Company]

Reviewers' comments:

Reviewer's Responses to Questions

**Comments to the Author**

1. Is the manuscript technically sound, and do the data support the conclusions?

Reviewer #1: Partly

2. Has the statistical analysis been performed appropriately and rigorously? 

Reviewer #1: Yes

3. Have the authors made all data underlying the findings in their manuscript fully available?

Reviewer #1: Yes

4. Is the manuscript presented in an intelligible fashion and written in standard English?

Reviewer #1: Yes

5. Review Comments to the Author

Reviewer #1: The manuscript by Tamanaha et al addresses an important aspect of the COVID-19 pandemic namely the tolerance of current molecular diagnostic techniques, in this case RT-LAMP, to emerging variants. However, there are a number of concerns that greatly reduce the potential of this manuscript that should be addressed before publication:

1. It would be much more informative if a comparative study was carried out between LAMP with the current gold standard i.e. qRT-PCR. The only references given for the supposed effect of mutations on qRT-PCR comes for references from 1989 and 2001 which were themselves praising another competing technology i.e. ARMS. The major message of the manuscript is that LAMP is more robust than qRT-PCR, yet there is no comparative evidence presented to prove that this is really the case.

2. Furthermore, the main experimental evidence given in this manuscript is a series single base mutations yet in the introduction it admits that emerging strains contain multiple mutations and frequent deletions although these scenarios were not considered in this manuscript. Mutations in at least 2 or 3 positions and simple deletions should be considered.

3. Whilst the alpha variant deletion was considered experimentally the authors admitted that LAMP was unable detect this deletion and instead used a molecular beacon to identify this deletion. The use of molecular beacon in an assay however greatly reduces the advantage of using LAMP in the first place, however no mention was made of this short-coming.

4. It is not clear why the authors chose to use synthetic SARS-CoV-2 RNA spiked into RNA from Jurkat, a T-cell line with without considering the effect of the experiments on clinical samples or at least nasopharyngeal or saliva samples.

5. Taking in consideration the points mentioned above the conclusions have been drown from an overinterpretation of the data rather than being fully supported by them.

6. PLOS authors have the option to publish the peer review history of their article (what does this mean?). If published, this will include your full peer review and any attached files.

Reviewer #1: No

---

## [Author Response · Author response to Decision Letter 0]

1 Feb 2022

Reviewer Comments and Author Responses:

Reviewer: It would be much more informative if a comparative study was carried out between LAMP with the current gold standard i.e. qRT-PCR. The only references given for the supposed effect of mutations on qRT-PCR comes for references from 1989 and 2001 which were themselves praising another competing technology i.e. ARMS. The major message of the manuscript is that LAMP is more robust than qRT-PCR, yet there is no comparative evidence presented to prove that this is really the case.

Authors: We agree with the reviewer that a comparison against PCR would be informative and considered conducting a similar evaluation with a PCR assay. This would of course take significant time and resources, 44 oligonucleotide primers and 24 dual-labeled probe oligos tested in ~1200 reactions for just one of the Gene N primer+probe sets in our SARS-CoV-2 qPCR kit, for example. The older citations we found relevant to the effects of mismatch positioning on qPCR but we did have references to several SARS-CoV-2 qPCR tests where single-base mutations caused significant impact. This was obviously not clear in the original draft, and in the interim months since our submission several others have been published all showing a significant impact of single-base mutations. We have updated the reference list for these papers (now Refs. 1-12) and moved and updated the relevant text in the introduction, which now reads as below. We feel this is a well-established finding from many other groups and is now better presented alongside our RT-LAMP results. 

Updated text:

The effects of mutations on qPCR assays are well-studied, with even single-base mutations having generally deleterious effect (1-8, 10, 11). Likewise, single-base changes from SARS-CoV-2 mutations have been demonstrated to have significant impact on RT-qPCR assays. One N gene mutation (G29140U) falling in a Forward PCR primer resulted in a 5–6 Cq increase with an estimated 67-fold drop in assay sensitivity (12). Another N gene single base change (G29179U) in the CDC N2 Forward primer produced a ~4 Cq increase in PCR and a ~5-fold decrease in sensitivity (13). A study of 14 SARS-CoV-2 RT-qPCR assays with mismatches identified several primer positions as particularly sensitive to mutation, with up to a 7.6 Cq increase compared to fully base-paired templates (7). 

Reviewer: Furthermore, the main experimental evidence given in this manuscript is a series single base mutations yet in the introduction it admits that emerging strains contain multiple mutations and frequent deletions although these scenarios were not considered in this manuscript. Mutations in at least 2 or 3 positions and simple deletions should be considered.

Authors: The reviewer is certainly correct that our study did not consider all potential mutation patterns. However, the most common occurrence is a single-base mutation affecting the primer binding region, as described in the PCR data above, our monitoring of mutations with the PrimerMonitor tool, and the only to-date variant mutation in a LAMP primer set, found in Gene E with the Omicron variant. Based on the >5M SARS-CoV-2 sequences compared against the primers to date none feature >1 mutation in the primer regions likely due to just the probability of relatively rare mutation frequency occurring >1 time in a short sequence span. And while we would of course like to have data for the effects of 2 or 3 mutations, that becomes extremely difficult to profile comprehensively as we have done here. For example, in just the As1e primer set there are 195 positions represented. To test all combinations of 2 mutations in that set would require 18,195 primer sets with >50,000 LAMP reactions, just for the one primer set. 

We do agree that profiling deletions would be useful as well, and have added experimental data to this point. Testing deletions at all positions would require generating RNA templates with deletions at all positions rather than the order of oligos, a difficult and expensive project to undertake, but we rely on the naturally occurring deletion we have already discussed. In earlier unpublished work we had attempted to target the described SGF deletion using primer design alone, placing the 9-base deletion at the very end of the FIP primer. Unfortunately, this resulted in a partially affected assay, reducing sensitivity and speed, but not providing a sufficient design for variant discrimination as we had intended. But now based on this reviewer point we have added this data into the manuscript, showing that RT-LAMP tolerates even large stretches of mismatches in the most sensitive primer position, reflecting a real-world “worst-case scenario” where the assay is affected but still amplifies. This now builds into the last section describing the variant calling using molecular beacons. As a result, this is now a more complete picture of mutation tolerance. Updated and new text for these points is below:

Previous LAMP assays have been designed specifically to target single nucleotide polymorphisms (SNPs) in a sequence by placing the mutation directly at the 3′ or 5′ terminal base of both the FIP and BIP primers (28). However, this arrangement is intended to target a known mutation and would not occur in a standard LAMP diagnostic assay. Natural mutations would most frequently consist of single base changes at any location along the LAMP primers, or more rarely, multi-base deletions. For example, the B.1.1.7 Alpha variant, in addition to the 6-base deletion in the S gene described above, contains a 9-base deletion removing 3 amino acids in Orf1a (3675–3677, the “SGF” deletion), and also >18 characteristic single base change mutations. The B.1.617.2 Delta variant contains an additional >20 different point mutations and 2 deletions whileOmicron (B.1.1.529) contains >30 point mutations and 4 deletions unique from both the Alpha and Delta variants (29). 

Emerging SARS-CoV-2 sequence changes can be monitored for potential overlap with amplification assay primers through New England Biolabs Primer Monitor tool. Using this tool, mapping common RT-LAMP primers onto SARS-CoV-2 sequences from GISAID shows numerous locations of mutation in small numbers of isolates worldwide. Of higher concern are mutations seen with significant prevalence, (>10% of sequences, deposited from at least 2 geographic locations, independent from any of the well-known variants of interest or concern; see Materials and Methods). In each case, only a single base change occurred within any of the primer set regions. In this study, we sought to comprehensively characterize the impact of mutations on RT-LAMP primers, and transferred the prevalent mutations to each position of all 18 oligonucleotide primers making up 3 SARS-CoV-2 RT-LAMP assays. As shown below, RT-LAMP displayed consistent performance in speed and sensitivity across all 523 mutational positions tested. 

While this robustness is valuable for a reliable, variant-tolerant molecular diagnostic assay, there is additional value in assays that can distinguish and specifically identify particular variants with the amplification reaction. Sequencing of positive samples will remain the most powerful and sensitive method for variant identification, but molecular diagnostic assays can present a significantly faster and cheaper approach, if appropriately designed. The widely-used TaqMan probes rely on specific hybridization to targeted sequences and may display greater sensitivity to mutations than when they occur in primer regions; the Δ69-70 deletion occurs in the TaqPath probe region and completely prevents detection of RNA with that deletion. While only Alpha and Omicron carry the Δ69-70 deletion, all but Delta carry a 3-amino acid deletion at positions 3675–3677 (3674–3676 in Omicron) in the Orf1a sequence. This relatively large deletion provides a reliable means for designing molecular diagnostic assays that can distinguish between strains differing at this sequence location (8), and we sought to target that region with deletion-specific LAMP primers. While there was some impact on speed and sensitivity, even this significant stretch of mismatches did not inhibit LAMP completely and showed little effect at high copy numbers. Instead, we show below a molecular beacon strategy is needed for variant targeting using RT-LAMP. 

Effect of Short Deletions on RT-LAMP and Detection by Molecular Beacons 

We next looked to characterize the effects of short deletions on RT-LAMP, as have occurred in several SARS-CoV-2 assays. Systematically screening positional deletion effects, as done with single-base mutations above, would require the generation of hundreds of RNA templates. Instead, we took the existing, frequently occurring Orf1a SGF deletion and designed primer sets with that mutation at the 5′ end of the FIP (Table 1), the position of highest sensitivity to mismatches, as determined above. At the the early phase of LAMP amplification, this FIP forms a mismatched 3’ end with wild-type target in the dumbbell. This multi-base mismatch would then interfere with the amplification from the dumbbell and also subsequent priming for polymerization. Testing RT-LAMP containing this SGF-Del-FIP with low copy number (~10 copies), amplification with fully matched SGF deletion target (B.1.1.7 RNA) was efficient and sensitive while no amplification was observed with the mismatched wild-type target, indicating that mismatched FIP affected detection (S3 Fig). However, with higher copy numbers (~1000 copies), primer sets with mismatched target sequence were not completely discriminatory, with amplification of B.1.1.7 RNA and wild-type primers (and vice versa), though with reduced speed (S3 Fig). These results indicates that while some degree of LAMP impact was achieved by placing the SGF deletion at the 5′ end of the FIP, the effects on amplification were insufficient for confident and consistent variant identification with this strategy. And while robust to even 9-base deletions at the FIP end with moderate copy numbers, the significant impact in low copy reactions would indicate mutations of this severity at key positions and would affect the performance of a LAMP diagnostics assay. 

Reviewer: Whilst the alpha variant deletion was considered experimentally the authors admitted that LAMP was unable detect this deletion and instead used a molecular beacon to identify this deletion. The use of molecular beacon in an assay however greatly reduces the advantage of using LAMP in the first place, however no mention was made of this short-coming.

Authors: The reviewer is correct that using fluorescence detection with the molecular beacon does indeed detract a bit from the simplicity advantages of LAMP. However, we only use the beacon for variant identification, a specific use case that we wanted to show was indeed possible with the otherwise variant-tolerant LAMP method. To make this clearer we have added the text below to the Discussion:

Requiring a molecular beacon and fluorescence detection does, of course, remove some of LAMP’s advantages in portability and simplicity, but in situations where variant calling is desired, LAMP can enable these applications, still with faster reaction times than PCR and on plate readers and simpler detection equipment that does not require thermal cycling.

Reviewer: It is not clear why the authors chose to use synthetic SARS-CoV-2 RNA spiked into RNA from Jurkat, a T-cell line with without considering the effect of the experiments on clinical samples or at least nasopharyngeal or saliva samples.

Authors: Use of clinical material would certainly be good and is, of course, necessary for validating a potential clinical assay, but obtaining consistent sample material for the 5,000 LAMP reactions we show here would have been very challenging especially as we wanted to ensure quantitation and equivalence across the various experiments. We store the low-concentration SARS-CoV-2 RNA in a background of Jurkat RNA simply for stabilization as is a common practice for low-copy nucleic acids. Scores of papers and studies have now looked at extracted RNA from clinical samples without any reason to suspect that material would behave differently in examining the effects of specific sequence changes on primer efficiency. 

Reviewer: Taking in consideration the points mentioned above the conclusions have been drown from an overinterpretation of the data rather than being fully supported by them.

Authors: We have extensively edited the manuscript to address the above concerns, added additional context and data to the study, and accordingly feel our conclusions are now more solidly supported by the data we show. We hope the reviewer agrees with this in our updated and revised manuscript.

---

## [Editor Report · Decision Letter 1]

1 Mar 2022

Profiling RT-LAMP tolerance of sequence variation for SARS-CoV-2 RNA detection

PONE-D-21-33500R1

Dear Dr. Tanner,

We’re pleased to inform you that your manuscript has been judged scientifically suitable for publication and will be formally accepted for publication once it meets all outstanding technical requirements.

Kind regards,

Grzegorz Woźniakowski, Full professor, PhD, ScD

Academic Editor

PLOS ONE
---

## [Editor Report · Acceptance letter]

15 Mar 2022

PONE-D-21-33500R1 

Profiling RT-LAMP tolerance of sequence variation for SARS-CoV-2 RNA detection 

Dear Dr. Tanner:

I'm pleased to inform you that your manuscript has been deemed suitable for publication in PLOS ONE. Congratulations! Your manuscript is now with our production department. 

Kind regards, 

on behalf of

Prof. Grzegorz Woźniakowski 

Academic Editor

PLOS ONE